# Spatial Fingerprinting: Horizontal Fusion of Multi-Dimensional Bio-Tracers as Solution to Global Food Provenance Problems

**DOI:** 10.3390/foods10040717

**Published:** 2021-03-28

**Authors:** Kevin Cazelles, Tyler Stephen Zemlak, Marie Gutgesell, Emelia Myles-Gonzalez, Robert Hanner, Kevin Shear McCann

**Affiliations:** Department of Integrative Biology, University of Guelph, Guelph, ON N1G 2W1, Canada; tzemlak@uoguelph.ca (T.S.Z.); mgutgese@uoguelph.ca (M.G.); emelia.myles-gonzalez@tulloch.ca (E.M.-G.); rhanner@uoguelph.ca (R.H.); ksmccann@uoguelph.ca (K.S.M.)

**Keywords:** food provenance, species origin, bio-tracers, data fusion, supervised learning

## Abstract

Building the capacity of efficiently determining the provenance of food products represents a crucial step towards the sustainability of the global food system. Despite species specific empirical examples of multi-tracer approaches to provenance, the precise benefit and efficacy of multi-tracers remains poorly understood. Here we show why, and when, data fusion of bio-tracers is an extremely powerful technique for geographical provenance discrimination. Specifically, we show using extensive simulations how, and under what conditions, geographical relationships between bio-tracers (e.g., spatial covariance) can act like a spatial fingerprint, in many naturally occurring applications likely allowing rapid identification with limited data. To highlight the theory, we outline several statistic methodologies, including artificial intelligence, and apply these methodologies as a proof of concept to a limited data set of 90 individuals of highly mobile Sockeye salmon that originate from 3 different areas. Using 17 measured bio-tracers, we demonstrate that increasing combined bio-tracers results in stronger discriminatory power. We argue such applications likely even work for such highly mobile and critical fisheries as tuna.

## 1. Introduction

Today’s global food system is a collection of highly inter-connected trade networks that span a myriad of organizations and geographies [1,2,3,4]. While such a system represents an important and perhaps necessary mechanism for meeting demands for nutritious and affordable food [3,5], it also represents a complex web of activity that carries with it a number of inherent challenges such as sustainability and transparency [2,6]. Most food items travel thousands of kilometers [7] changing form and ownership several times before reaching a consumer’s plate [4]. Without proper labelling—such as Country of Origin Labelling (COOL) regulation—the consumer does not have the capability to accurately identify where their food originated and thus cannot make informed decisions about the products they are buying [8]. Unfortunately, tracing food commodities back to their respective origin is a formidable task, which can only be tackled by a robust traceability system integrated along the entire food supply chain [9].

The introduction of new technology (e.g., Wireless Sensor Network and Radio Frequency IDentification, blockchain) and ad hoc recommendations(ISO 9000, Codex Alimentarius, etc.) represent indispensable tools to monitor and secure different food chain stages [9,10,11]. However, there is one common limitation that is stopping us from realizing robust provenance-based value chains—the ability to verify traceability information [12]. Consequently, a vigorous research effort has been geared towards the development of methods to authenticate type and origin of food commodities, such as sensory analysis and chromatographic techniques [12,13,14]. One promising avenue is the use of bio-tracers, i.e., biological features (e.g., DNA, trace-elements, metabolomic compounds, stable isotopes, etc.) that vary with (and thus reflect) the environment an individual is living in, to create fingerprints that can recognize different food products. For instance, DNA barcodes have been used for over a decade to uncover fraudulent labelling practices in the seafood industry [15,16,17,18,19]. Similarly, stable isotopes have shown a lot of promise for authenticating the origins of various food products including olive oils, cheese, honey, meat and fish [20,21,22].

In food product authentication, classes of bio-tracers are often employed independently of each other and “vertical” bio-tracer strategies (i.e., using different markers within a given class of bio-tracers) still prevail to adjust the granularity of information being sought. For example, small DNA sequence fragments of the mitochondrial cytochrome c oxidase I gene (COX-1) are enough to identify a fish fillet to species [19], but the genome coverage required is larger when trying to discriminate sub-populations where genetic variability is much smaller [17]. Similarly, increasing the number of stable isotopes used has repeatedly been shown to be a powerful approach to determine the provenance (i.e., the origin) of numerous food products, even at fine spatial scales [21,23,24,25,26]. Interestingly, despite the evidence of important gains in discriminatory power brought by vertical data fusion, the general reasons behind such success are rarely discussed. Furthermore, it remains unclear whether this gain extends beyond one class of bio-tracers, i.e., the potential of “horizontal” strategies for food authentication remains to be established [12,27].

In what follows we discuss the efficiency of combining information from different bio-tracers (vertically and horizontally) for food authentication with a specific focus on provenance. Our goal here is threefold. First, we explain how to use multiple bio-tracers to create spatial fingerprints. Second, we show that increasing the number of bio-tracers for authentication increases authentication performance. Third, we provide a relatively simple explanation for why data fusion is always a winning strategy and comment on the potential of horizontal strategies. We support our arguments (which are mainly mathematical, see Appendix A) by comparing how well a set of bio-tracers perform when trying to assign Sockeye Salmon (*Oncorhynchus nerka*) to three geographically distinct fisheries: British Columbia, Canada; Kamchatka Peninsula, Russia; and Alaska, United States (Figure 1). The set of bio-tracers included three isotopes ($\delta^{15}$N, $\delta^{13}$C and $\delta^{34}$S) and 14 fatty acids for a total of 17 bio-tracers spanning two different classes. The Sockeye fishery itself presents an interesting model because sustainability practices vary somewhat geographically. The entire Alaskan fishery is certified by the Marine Stewardship Council (MSC)—the most rigorous and widely recognized eco-certification available. The Canadian fishery was also recognized by MSC as sustainable, until 2019 when the Canadian Pacific Sustainable Fisheries Society (CPSFS) decided to self-suspended its MSC certification for many salmon species, including Sockeye [28]. While some fisheries in Russia received certification from the MSC, many remote fisheries in Eastern Russia are under threat due to extractive industries, loss of habitat and large-scale poaching [29]. Much of this is thought to be driven by linkages to organized crime in east Asian markets [30,31]. Therefore, building the capacity to distinguish high-level geographic origins of Sockeye is of particular relevance to the sustainability of Sockeye fishery and food provenance interests in general.

## 2. Materials and Methods

### 2.1. Data

Muscle tissue trimmings were collected from 90 Sockeye salmon individuals from three different regions (30 individuals per region): British Columbia, Canada; Kamchatka Peninsula, Russia; and Alaska, United States were donated by Albion Farms & Fisheries Ltd. (now Intercity Packers Ltd.), Richmond, BC, Canada. All samples were derived from fillet trimmings to simulate a likely Quality Assurance/Quality Control scenario. Each muscle trimming was processed to obtain 2 muscle tissue samples for analyzing 17 bio-tracers of two classes: 3 stables isotopes (δ^15^N, δ^13^C and δ^34^S) and 14 fatty acids (C16:0, C16:1, C18:0, C18:1, C18:2n-6, C18:2n-6, C18:3n-3, C18:4n-3, C20:1, C20:4n-3, C20:5n-3, C22:1, C22:5n-3, C22:6n-3 and C24:1). One muscle sample from each fish was delivered frozen to the Lipid Analytical Services at the University of Guelph for fatty acid analysis using a combination of Bligh and Dwyer and Morrison and Smith methods [32,33]. Individual FA weights (μg/g) were converted to a % FA composition and fatty acids with >1% presence were retained as bio-tracers. The second muscle samples were dried at 70 ${}^{\circ }$C for 2 days and ground into a fine powder in preparation for stable isotope analysis. Tissue samples were sent to the University of Windsor GLIER Chemical Tracers Lab for isotopic analysis of δ^15^N, δ^13^C and δ^34^S (Windsor, ON, Canada). Importantly, all variables were centered and scaled before any statistical inference.

### 2.2. Numerical Simulations

#### 2.2.1. Statistical Models

We created spatial fingerprints of increasing complexity by combining up to 17 bio-tracers for our three regions of interest (see Figure 1) and then evaluating their performances (on a different set of samples) to correctly determine the origin of a sample (see the following section). Among the large diversity of supervised-learning methods available, we chose three to reflect current and emerging practices in food authentication:Linear Discriminant Analysis [34] (LDA), see Sun et al. [35] for a use case;Naive Bayesian Classifier [34] (NBC), see Wunder [36] and Bataille and Bowen [37] for examples;A Multi-Layer Perceptron [34] (MLP), see Wu et al. [27] for a recent study.

For all three methods, we assessed the probabilities of correctly assigning a sample to its true origin (referred to as *performance*) for every region (this corresponds to the diagonal of the confusion matrix) as well as the probability of assigning a sample to its true origin, irrespective of its true provenance (overall performance). Assuming that we have no prior expectation for the origin of a given sample, the overall performance corresponds to the mean of the diagonal of the confusion matrix.

#### 2.2.2. Simulations Design

For every simulation, we randomly selected 20 samples per regions (60 samples total) as training set and used the remaining samples (10 per region) to evaluate performances of combinations of bio-tracers (thus, the samples used to evaluate performances are different from the one use by the algorithm to create its own knowledge of the data). All simulations were replicated for all three selected classification approaches. We also evaluated the impact of respective size of the two data sets for and we show that gains of performances beyond 20 samples in the training set were marginal for all three methods (see Figure A8).

We evaluated the performances for an increasing number of bio-tracers (from single performances up to the combination of all the 17 bio-tracers available). For every number *p* of bio-tracers, we used 500 combinations of *p* bio-tracers. When there were less than 500 existing combinations, we used all of them. For every combination, we randomly chose 200 pairs of training and test sets, leading to up to 100,000 simulations for a given number of bio-tracers. We also assessed the overall performances of the three approaches on the dataset ordered by a Principal Component Analysis (PCA). PCA is a statistical tool commonly used to reduce dimension [38], here PCA was used to transform our data set and obtained uncorrelated variables ordered according to the percentage of variance of the entire data set they capture. To evaluate the robustness to noise, we added an increasing amount of white noise in the of the training set, i.e., for every simulation, we drew 60 values in a centered normal distribution of an increasing standard deviation (from 0.0001 to 10). For every simulation, we used 500 combinations of bio-tracers and 200 pairs of training and test sets (randomly chosen).

Finally, for all bio-tracers and all combinations of 2 and 3 bio-tracers, we computed the inter-regions variance as well as the distance between region centroids (coordinates of region centroids are the means of coordinates of all samples in a given region). We also computed the region data overlap. To do so, for the three regions studied, we computed the convex hull for all pairs and triplets of bio-tracers. Note that, in order to discard potential outliers, we only used 27 data points per region (90%), points included were the closest to their respective region centroid. We then computed the volume (or area) of all intersections between the three convex hulls, summed them and then divided the quantity thereby obtained by the total area (or volume) of the three convex hulls. Last, for all of these sets of bio-tracers we evaluated the performance bio-tracers using 1000 pairs of training and test sets (randomly chosen).

### 2.3. Mathematical Proof

In Appendix A, using Bayes’s rule, we demonstrated that increasing the number of bio-tracers combined almost surely increases the discriminatory power (performance) of a Naive Bayesian Classifier (NBC).

#### Numerical Implementation

For LDA, we used the R implementation *lda()* available in the package “MASS” [39]. We implemented our own naive Bayesian classifier using R version 3.6.3 [40] and use the function *density()* for kernel density estimates.

Finally, we used the Julia library Flux.jl [41] for the multi-layer perceptron (two dense layers and cross-entropy loss function). As this approach is data demanding, we used a simple data augmentation procedure: data in the training set were repeated and noise (random variables drew from a centered normal distribution of standard deviation σ) of various levels was added to it (as a centered normal distribution). After evaluating the performances under various augmentation scenarios (see Figure A6), we opted for 1000 repetitions of the data set and a noise level of σ=0.01.

## 3. Results

### 3.1. The More Bio-Tracers the Better

For the three regions considered, increasing the number of bio-tracers always increased the probability of correctly assigning a sample to its true origin (Figure 2). The three statistical approaches considered show similar behavior, qualitatively, with MLP having the best performance (Figure 2c). All three methods consistently exceed 90% of correct assignment when 12 or more bio-tracers are combined for Canadian and Russian samples. The three approaches also perform significantly less efficiently for Alaskan samples, which are geographically closer to the two other regions Figure 1. Interestingly, the same order applied in the data space: the distance between Russia and Canada (based on the Euclidean distance between group centroids) is the longest (4.819 vs. 4.145 for Canada-USA and 2.536 for Russia-USA).

The overall performance (i.e, the probability of correctly determining the provenance of a sample irrespective of its true origin), based on a single sample, from 1 to 17 bio-tracers increases from 0.444 to 0.898 for LDA, from 0.465 to 0.817 for NBC and from 0.482 to 0.915 for MLP (Figure 2d–f). Moreover, performances are strongly improved when testing multiple individuals (Figure 2d–f). It is worth noting that even in such case, employing more bio-tracers still provides more accurate predictions (Figure 2d–f). Note that these results align fully with our analytical derivations (see Appendix A and Figure A3). Furthermore, increasing bio-tracers is very robust to noise addition, and this holds true for all three methods (Figure 3). For instance, the overall performance of LDA with 5 bio-tracers and a very low level of noise added (10${}^{-4}$) is 0.710, but combining 15 bio-tracers with an addition of a noise with a level as high as 1 (a fairly strong noise addition) still yields better discriminatory power (0.758). Therefore, even if the measurement are known to be less accurate for some bio-tracers, they are likely worth being combined with others, assuming that the error is consistent among samples.

Using the first axes provided by Principal Component Analysis (PCA) applied on the data set (see Methods) is a strategy that performs relatively well: across all three approaches, using up to the first 6 principal component axes is consistently better than the median of all the bio-tracer combinations we tested (Figure 4). Furthermore, as expected, the results obtained are similar when most or all bio-tracers are being used, except for NBC for which the PCA slightly negatively impacts the overall performance. Most importantly, for all methods, the axis order provided by a PCA (the first axis being the one that captures the most variance) does not necessary reflect their discriminate power. Hence, the three statistical methods show that the 5th principal component axis provides a more important gain in performance than the 4th one (Figure 4). In general, combining only a few of the first principal component axes to authenticate food products, as frequently done [42], may be a sub-optimal approach as it can discard axes that carry less variance but more discriminatory power.

### 3.2. An Examination of the Performances

Individually, the 17 bio-tracers have contrasting authentication performances (Figure 5), this holds true for both classes of bio-tracers: δ^15^N and oleic acid (C18:1) alone perform well (0.547 and 0.548, respectively) whereas δ^13^C and linoleic acid (C18:2n-6) perform poorly (0.343 and 0.333). It is worth noting that the top 3 bio-tracers, based on individual performances, includes 2 fatty acids (oleic acid and docosapentaenoic acid, i.e., C22:5n-3) and one stable isotope (δ^15^N; see Figure 5) and thus cover the two classes of bio-tracers. Note that even though we only show this for LDA (Figure 5), this holds true for NBC (see Figure A8 in Appendix B) and MLP (see Figure A11).

Interestingly, the overall performance of a pair of bio-tracers systematically outcompetes the best performing of the two bio-tracers included in the pair (see Figure 6a for the results for LDA and Figure A9a for NBC and Figure A12a for MLP). Similarly, the performance of combining the three bio-tracers is better than the best performing pair of bio-tracers that can be drawn from the triplet (see Figure 6b, Figure A9b and Figure A12b). Furthermore, the overall performance of a set of bio-tracers positively correlates with the performances of its subsets. Therefore using the best performing bio-tracers frequently yields a stronger discriminatory power (Figure 6c,d, Figure A9c,d and Figure A12c,d). This explains that the top 3 bio-tracers, and thus the two classes of bio-tracers, are systematically included in the best pairs and triplets (see Table A1 and Table A2).

As expected, the percentage of inter-regions variance captured by a bio-tracer is strongly and positively correlated with its overall performance (Figure 5a and Figure 7a,b). Even in 2 and 3 dimensions, simple non-linear least-squares regressions efficiently captures the variance of these relationship (R^2^ = 72.0% and 48.1% for LDA, respectively, see Figure 7a,b, Figure A10a,b and Figure A13a,b for NBC and MLP, respectively). In one dimension, the mean Euclidean distance between region centroids efficiently summarizes one key geometrical results of the data space: the further apart the data points of different regions are, the stronger the discriminatory power (Figure 5b). This result could be seen as a simple case of a more general one: the less overlap among regional hypervolumes (i.e., hypervolumes generated by data points of the different regions), the stronger the discriminatory power of a set of bio-tracers gets (Figure 7c, Figure A10c and Figure A13c). Notably, increasing dimensions is often an efficient way to reduce overlap among regions data points (see Figure 7c, Appendix A, Figure A10c and Figure A13c).

## 4. Discussion

Working in high dimensions for reliable authentication is already being used in our day-to-day life. For instance, face recognition algorithms use a high number of “abstract features” to recognize faces [43,44]. Similarly, multi-messenger astronomy is experimenting with the fusion of electromagnetic radiation, gravitational waves, neutrinos and cosmic rays to observe and understand the universe through a new lens [45]. Here we acknowledge similar potential for food authentication and clarify why data fusion can enhance the discriminatory power of traceability tools, and thus play a major role in food authentication in the foreseeable future, as other authors have predicted [12]. Our simulations suggest that multi-tracer approaches are increasingly strengthened by spatial tracer covariance and, importantly, allow rapid provenance detection even in the face of noise relative to low dimensional approaches. This is especially relevant for horizontal data fusion of bio-tracers as they are plentiful—some of which have just started to reveal their potential in tracing food products [46]—and reflect various interactions between individuals and their immediate environment. Hence, together with technical advancements that trace movements of food products (such as blockchain), using bio-tracer based fingerprinting strategies to verify the origin of food product can contribute to making the food supply chain more transparent, more robust and eventually more sustainable.

Even though working in high dimension could be a very efficient approach, it also comes with its challenges: even if additional dimensions increase the discriminatory power of statistical classifiers, it comes at a cost as probability density estimates are more difficult and thus less accurate [38]. This is where dimension reduction methods, such as PCA, can be utilized, as they allow for working in a simpler space with mathematically-desirable properties (e.g., uncorrelated axes that concentrate the variance). However, one needs to bear in mind that what matters is to keep as much discriminatory power as possible and thus one should realize that, for instance, working with only the few first principal components may not always provide the best authentication tool as dimensions representing a low amount of total variance may still be of major importance to separate a pair of regions or more. Researchers should rather focus on statistical tools that reduce dimensions while maximizing discriminatory power, such as stepwise LDA [47]. Fortunately, the recent boom in artificial intelligence research is bringing considerable methodological advancements in multivariate density estimation and dimension reduction [48].

Taking advantage of data fusion can only be achieved if relentless efforts are made to acquire reliable data that would be securely archived (e.g., within a blockchain) while being widely accessible. This would require creating and maintaining ad hoc digital infrastructure. In our Sockeye example, we only needed 90 samples and 17 bio-tracers to cleanly differentiate a globally ubiquitous species by geographical region; however, we only covered 1 species across 3 spatially coarse regions—making high diversity and fine spatial scale applications will require more intensive data and probably the integration of additional classes of bio-tracers. Although we did not consider DNA approaches beyond COI barcoding, we did explore DNA barcodes, well known for its species identification abilities [49,50], as a tool for spatial identification. Nonetheless, the genotypic variation at the COI gene was small and showed no spatial signal (see Figure A14). Moreover, here we did not investigate the temporal variations in bio-tracers distribution for the different regions which will be a critical step as this would determine the survey frequency required to maintain reliable spatial fingerprints. Overall, the sampling effort and the data required to extensively cover fishing areas experiencing food security concerns with numerous species of interest (and/or at risk) over long periods of time would certainly be bigger by several orders of magnitude.

Ultimately, standardizing sampling protocols, building large databases and employing powerful computational tools will allow researchers and national authorities to create dynamic maps of probability of origin for any food product to be tested [37,51]. There are various strategies to improve food authentication, employing horizontal data fusion is clearly one of them. Fortunately, we are living in an era where major technical needs have been met, thus horizontal strategies can be employed immediately, but evidently their spread will depend on the balance between the cost of their application and the economical benefits for fishing industry, which vary across seafood products. That said, horizontal data fusion of bio-tracers could certainly be employed beyond the field of food authentication as it is a general principle where bio-tracers can be applied and combined to determine a wide array of biological properties, be it for determining the origin of a species or the structure of an entire food web.

## Figures and Tables

**Figure 1 foods-10-00717-f001:**
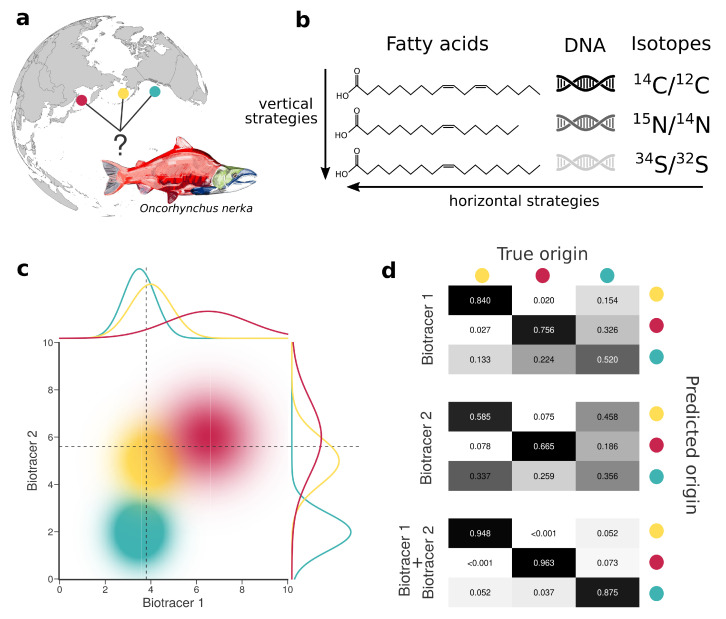
Combining bio-tracers to improve the determination of samples’ provenance. (**a**), Sockeye salmon (*Oncorhynchus nerka*) samples of this study originate from three potential origins, namely Alaska, United States (yellow); British Columbia, Canada (cyan) and Kamchatka Peninsula, Russia (magenta). (**b**), We examine the efficiency of horizontal strategies that combine several classes of bio-tracers as opposed to vertical strategies that focus on one specific class. (**c**), While using a single bio-tracer to discriminate the true origin of a sample (distributions on top and right of the chart, dotted line depict bio-tracer values of a sample) may prove difficult, combining bio-tracers (colored areas) enhances the performance of the inference process. (**d**), This is also shown with confusion matrices obtained using a classifier that uses only the first bio-tracers (top), only the second one (middle) or the combination of the two (bottom).

**Figure 2 foods-10-00717-f002:**
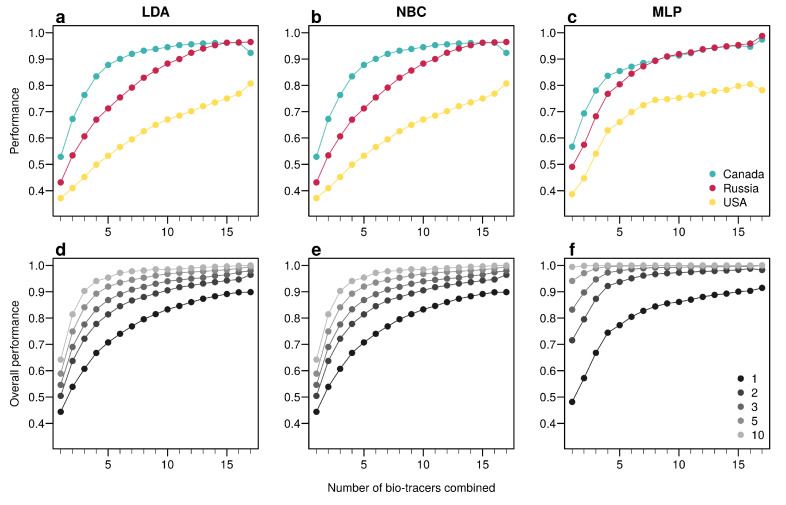
Increasing the number of bio-tracers considerably improves statistical performances. (**a**–**c**), The probability of assigning one sample to its true origin increases as the number of bio-tracers employed increases for the three regions considered, namely Alaska (yellow), British Columbia (cyan) and Kamchatka Peninsula (magenta). (**d**–**f**), The overall performance (i.e., the correct assigning any sample to its true origin) can also be improved by combining samples, assuming samples combined originate from the same region (e.g., individuals of the same lot). Points are colored according to the number of samples combined. These results are qualitatively similar for the three statistical approaches considered, which are Naive Bayesian classifier (NBC; (**a**,**d**)), Latent Discriminant Analysis (LDA; (**b**,**e**)) and a Multi-Layer Perceptron (MLP; (**c**,**f**)). In all panels, points represent performances averaged over up to 100,000 replicates (see Methods for further details).

**Figure 3 foods-10-00717-f003:**
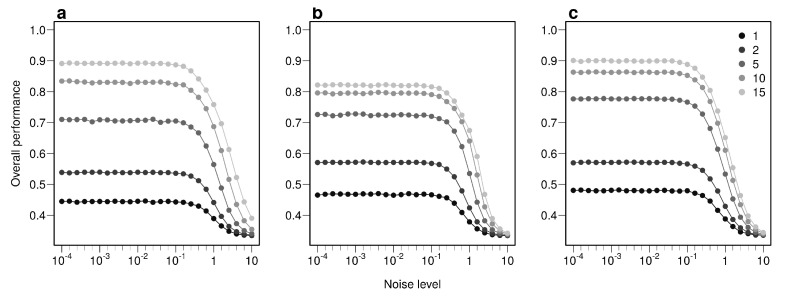
Combining bio-tracers is robust to noise addition. The probability of correctly determining the provenance of samples is evaluated for an increasing noise addition to the training data set. The lighter the gray, the more the number of bio-tracers combined. Note that prior to analysis, all bio-tracer values were scaled, thus a noise level of 1 represents a strong noise addition. The three panels correspond to three statistical approaches used: NBC (**a**), LDA (**b**), MLP (**c**).

**Figure 4 foods-10-00717-f004:**
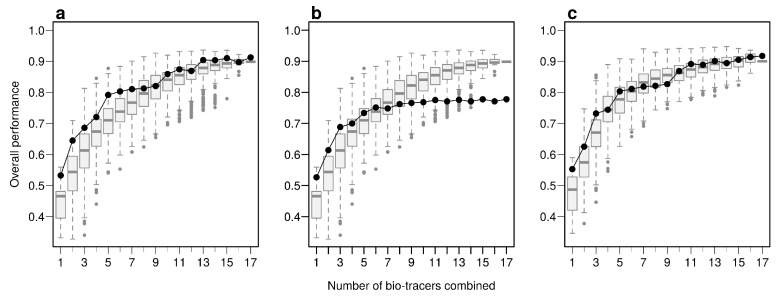
PCA does not necessarily provides axes with the maximum discriminatory power. Boxplots represent the probability of correctly determining the provenance of one sample for 500 combinations of bio-tracers (or all combinations if the total number of combinations is less than 500; see methods for details). Black lines and points represent results obtained when the first of principal component axes are being used. The three panels corresponds to three statistical approaches used: NBC (**a**), LDA (**b**), MLP (**c**).

**Figure 5 foods-10-00717-f005:**
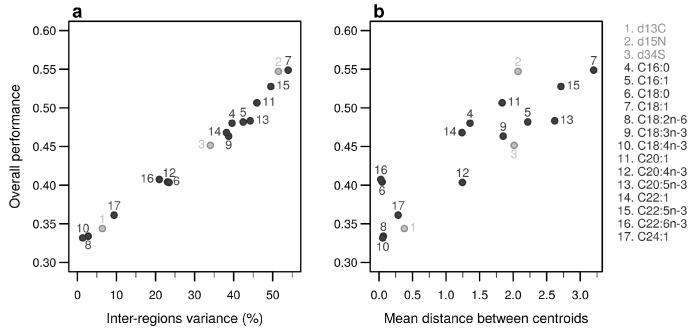
Performances of individual bio-tracers. Overall performances of all 17 bio-tracers (listed on the right) using LDA are plotted against the proportion of inter-regions variance (**a**) and the mean distance between all pairs of region centroids (**b**).

**Figure 6 foods-10-00717-f006:**
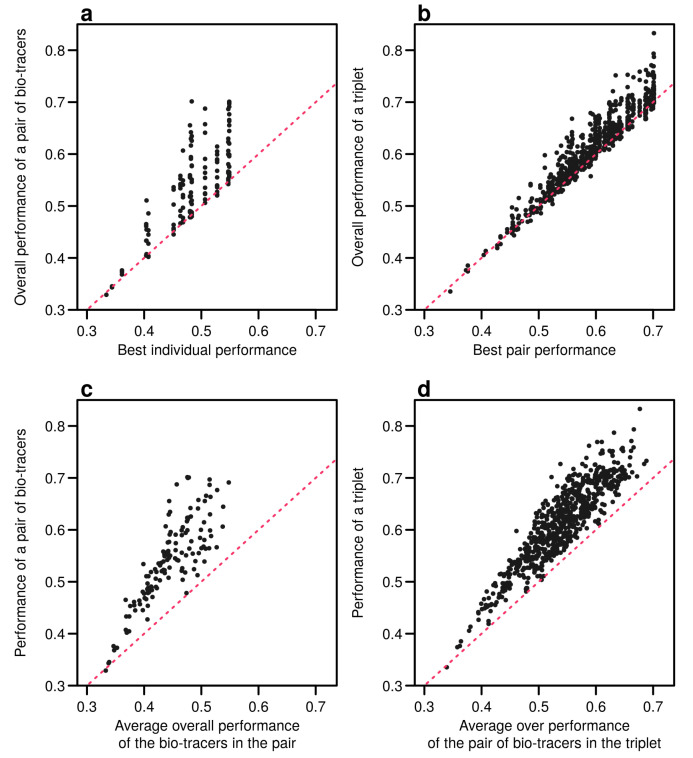
Including one more bio-tracers increases performance. Overall performances of all pairs of bio-tracers using LDA are plotted against the best individual performing bio-tracers of the pair (**a**) and their average overall performance (**c**). Similarly, overall performances of all triplets of bio-tracers are plotted against the best performing pair of bio-tracers of the triplet (**b**) and their average overall performance (**d**). Magenta dashed lines represent the 1:1 slope.

**Figure 7 foods-10-00717-f007:**
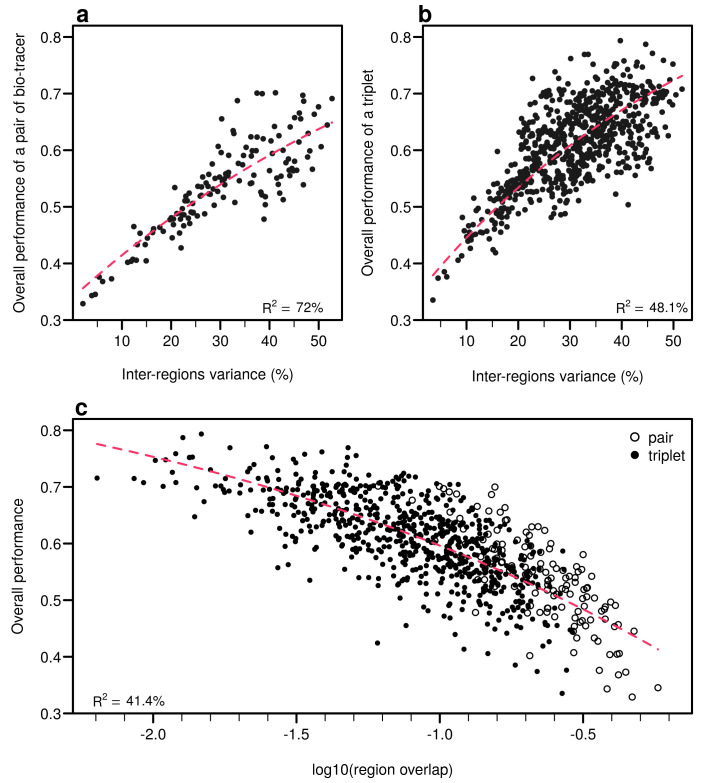
Efficient combinations of bio-tracers maximise inter-regional variance and minimize overlap of region data hypervolumes. (**a**,**b**) For all combinations of 2 (**a**) and 3 (**b**) bio-tracers, the overall performances (with LDA) of sets of bio-tracers are plotted against their inter-regions variance. (**c**) We present relationship between the proportion of overlap of data between regions and the overall performances for all pairs and triplets of bio-tracers. Magenta dashed lines represent the results of non-linear leas-squares regression and the corresponding R-squared are added at the bottom of every panel.

## Data Availability

Data are available from the Dryad Digital Repository at https://doi.org/10.5061/dryad.95x69p8jd, access on 13 March 2021. All code to replicate this study is archived as a research compendium on Zenodo at https://zenodo.org/badge/latestdoi/250544023, access on 13 March 2021.

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
