# Peer review of "Spatial Fingerprinting: Horizontal Fusion of Multi-Dimensional Bio-Tracers as Solution to Global Food Provenance Problems"

_foods, 2021, doi:10.3390/foods10040717_

Round 1
Reviewer 1 Report
The article entitled "Spatial fingerprinting: horizontal fusion of multi-dimensional bio-tracers as solution to global food provenance problems " presents a rigorous and structured work. It draws interesting results and conclusions. However, even in appendices, it includes too many mathematical formulae which are beyond the scope of this journal. The authors should reference books and scientific publications where interested readers can find these mathematical formulae.
For this reason I consider that the article needs major revision.
Author Response
The article entitled "Spatial fingerprinting: horizontal fusion of multi-dimensional bio-tracers as solution to global food provenance problems " presents a rigorous and structured work. It draws interesting results and conclusions.
We thank the reviewer for this comment.
However, even in appendices, it includes too many mathematical formulae which are beyond the scope of this journal. The authors should reference books and scientific publications where interested readers can find these mathematical formulae.
Mathematical derivations were done from first mathematical principles used in Bayesian probabilities (references have been added to point the reader to key papers). When advanced results were needed, references were added. Given the points made by reviewers 3 and 4, we decided to keep the appendice.
Reviewer 2 Report
Manuscript foods-1151319 reports on a strategic plan to determine the geographical origin of food products, focusing on seafood. The determination of the provenance of foodstuffs is a topic of great interest among researchers. The article is in total well written and organized and reports on evidenced chemometrics. However, the authors must improve the manuscript according to the following comments:
General Comments: Kindly avoid to use always the first person’’ we’’. Rephrase in the whole manuscript, Figure legends, etc.
-Introduction
-Line 31 and elsewhere: Correct the reference style in the text.
-Materials and Methods
Lines 166-167. These have some problems. What is the parenthesis used for?
-Results and discussion (or Introduction)
The previous studies that are similar, in terms of presenting strategies for food authentication using lower or larger sample size, must be included:
European Food Research and Technology, 245(4), 805-816 (2019).
Foods, 8, 210; doi:10.3390/foods8060210 (2019).
Based on the above, I suggest a major revision.
Author Response
Manuscript foods-1151319 reports on a strategic plan to determine the geographical origin of food products, focusing on seafood. The determination of the provenance of foodstuffs is a topic of great interest among researchers. The article is in total well written and organized and reports on evidenced chemometrics.
We thank the reviewer for this comment.
General Comments: Kindly avoid to use always the first person’’ we’’. Rephrase in the whole manuscript, Figure legends, etc.
We have been using "we" in a very long list of papers of our own. We would be happy to change it if there is a suitable and grammatically correct alternative.
Line 31 and elsewhere: Correct the reference style in the text.
Done.
Reviewer 3 Report
In the manuscript " Spatial fingerprinting: horizontal fusion of multi-dimensional bio-tracers as solution to global food provenance problems" by Kevin Cazelles et al., the authors discuss the efficiency of combining information from different bio-tracers (vertically and horizontally) for food authentication with specific focus on provenance based on three approaches: explaining how to use multiple bio- tracers to create spatial fingerprints; showing that increasing the number of bio-tracers for authentication increases authentication performance, and finally providing a relatively simple explanation for why data fusion is always a winning strategy and comment on the potential of horizontal strategies. The arguments are supported by comparing how well a set of bio-tracers perform when trying to assign Sockeye Salmon (Oncorhynchus nerka) to 3 geographically distinct fisheries
The work is technically sound and scientifically valid.
The used methodology is appropriate and the objectives of the work clearly defined, and the experimental design well planned. The conclusions drawn are fully supported by the data presented and the claims are appropriately discussed in the context of previous literature. In addition, the paper provided sufficient methodological detail that the experiments can be reproduced.
The introduction is adjusted and the results are well discussed.
A very interesting appendix is presented with several important mathematical formulations as a complement to better explain the models used on the results treatment.
Author Response
The work is technically sound and scientifically valid.
We thank the reviewer for all their positive comments.
Reviewer 4 Report
The manuscript entitled “Spatial fingerprinting: horizontal fusion of multi-dimensional bio-tracers as solution to global food provenance problems” discuss the efficiency of combining information from different bio-tracers (vertically and horizontally) for food authentication with a specific focus on provenance. In my opinion, this manuscript should be minor revisions.
Line 31: remove Galvez et al.
Line 282: please correct this information “see 282 Figure ??)”
Author Response
> The manuscript entitled “Spatial fingerprinting: horizontal fusion of multi-dimensional bio-tracers as solution to global food provenance problems” discuss the efficiency of combining information from different bio-tracers (vertically and horizontally) for food authentication with a specific focus on provenance. In my opinion, this manuscript should be minor revisions.
We thank the reviewer for this comment.
> Line 31: remove Galvez et al.
The correct reference is now used.
> Line 282: please correct this information “see 282 Figure ??)”
The correct reference to figure 14
Round 2
Reviewer 1 Report
Ok